# Maternal Social Hierarchy, Morphometric Traits, Live Weight, and Metabolic Status as Related to the Offspring Pre-Weaning Growth in Crossbred Dairy Goats

**DOI:** 10.3390/ani15081100

**Published:** 2025-04-10

**Authors:** Jessica Maria Flores-Salas, Ma Silvia Castillo-Zuñiga, Cesar Alberto Meza-Herrera, Ma Guadalupe Calderon-Leyva, Jorge Arturo Bustamante-Andrade, Ma de los Angeles de Santiago-Miramontes, Silvestre Moreno-Avalos, Alan Sebastian Alvarado-Espino, Viridiana Contreras-Villarreal, Francisco Gerardo Véliz-Deras

**Affiliations:** 1Unidad Laguna, Universidad Autónoma Agraria Antonio Narro, Periferico Raúl Lopez Sanchez, Saltillo 25315, Mexico; jesflor13@hotmail.com (J.M.F.-S.); gpecalderon06@gmail.com (M.G.C.-L.); angelesdesantiago867@gmail.com (M.d.l.A.d.S.-M.); smavalos91@hotmail.com (S.M.-A.); alanalvaes@gmail.com (A.S.A.-E.); 2Centro de Bachillerato Tecnologico Agropecuario N. 1, Torreón 27410, Mexico; mscz1970@hotmail.com; 3Unidad Regional Universitaria de Zonas Aridas, Universidad Autonoma Chapingo, Bermejillo, Durango 35230, Mexico; cmeza2020@hotmail.com; 4Facultad de Agricultura y Zootecnia, Universidad Juárez del Estado de Durango, Venecia, Durango 34113, Mexico; abaj_86@hotmail.com

**Keywords:** goats, social hierarchy, maternal status, pre-weaning growth

## Abstract

This study aimed to define the possible maternal social rank effect, either low or high, by quantifying diverse morpho-physiological maternal markers upon the birth-to-weaning growth dynamics of their kids. A behavioral study conducted 30 days before the expected kidding date defined the maternal social rank. The behavioral study was conducted during feeding time (i.e., 08:00, 13:00 and 17:00; 60 min test^−1^, 180 min d^−1^). Our research outcomes suggest that high-social-rank goats articulated physiological and morphometric responses to achieve better and greater access to food, increasing prepartum and postpartum live weight and zoometric values. However, these advantages were not reflected in a higher level of productivity regarding litter size or the kid or total offspring growth during the birth-to-weaning period. The last is an interesting physio-ethological strategy to assure the survival of their kids, irrespective of social rank, in crossbred dairy goats managed under dry–semi-arid marginal production systems.

## 1. Introduction

The goat (*Capra hircus* L.) is a highly gregarious species that lives in very stable social groups based on a linear social hierarchy of relationships among its members [1,2,3,4]. Despite the scientific attention this social hierarchy has received, its importance and dimension have not been fully explained; such information could be a key component for the welfare and management of animals in various production schemes [5,6]. This social hierarchy is a common characteristic in most domestic herbivores, especially in grazing systems, where food is available ad libitum. The dominant animal will generally have privileged access to better-quality resources than subordinate animals, regardless of greater food availability and diversity. Also, the food consumption of subordinate animals is usually disturbed by dominant animals, which can affect their physiological and morphological characteristics [7,8,9,10,11].

Different variables can modulate goat production, including physiological, ethological, and even morphological factors [12]. In this sense, dominant animals generally disturb the food consumption of subordinate animals, which can affect the mother’s and her offspring’s physiological and morphological characteristics [13]. Considering the importance of goat production, especially as a productive strategy to mitigate the effects of climate change, goats emerge as a recurring theme on the public agenda and scientific literature [14,15].

Based on the previous information, we hypothesize that female goats with a high social hierarchy have better physiological and morphometric values than subordinate goats. This translates into a better nutritional environment, generating higher offspring birth weights from this social rank. Therefore, we aimed to identify the possible effect of maternal social rank on various morpho-physiological indicators such as live weight, body condition, thoracic diameter, thoracic perimeter, and serum glucose content, to evaluate the possible maternal effect of such variables concerning the pre-weaning growth rate dynamics of their offspring.

## 2. Materials and Methods

### 2.1. Ethical Note

In the present investigation, animal management followed the technical specifications for producing, caring for, and using laboratory animals [16]. Moreover, all the experimental units considered in this research, as well as the experimental procedures, methods, and management, were developed considering the requirements issued for good ethical use, care, and animal welfare both at international [17] and national [18] levels and with institutional approval registered under the reference UAAAN-UL-38111-425501002-2837.

### 2.2. Location of the Study Area, Animals, and Management

The present study was carried out between October and December 2021 in northern Mexico, in the Comarca Lagunera (25°51′ N, 103°16′ W, 1190 m) located in the subtropics with a dry–hot semi-arid climate and an average annual rainfall of 266 mm (range: 163 to 504 mm; Jun to Sep) [19]. In this region, the photoperiod during the summer solstice is 13:41 h, and 10:19 h for the winter solstice. Crossbred dairy adult goats (Alpine–Saanen–Nubian × Criollo; *n* = 15, 2–3 yr. old, and managed under semi-intensive conditions where goats grazed from 10:00 to 18:00 h in different grazing sites on a path of approximately 6 to 8 km) were assigned to two groups based on their social rank (HSR, *n* = 8; LSR, *n* = 7). The onset of the study was 1 month before kidding, with Nov-25 as the expected kidding date. All goats were housed in a 150 m^2^ (10 × 15 m) pen for one month before kidding. The goats were fed alfalfa hay with free access three times a day (08:00, 13:00, and 17:00); water and mineral salts were provided similarly, considering the NRC requirements [20], and the kids were kept with their mothers and allowed to suckle freely.

### 2.3. Behavioral Study to Define the Maternal Social Rank

All goats (*n* = 15) were subjected to a behavioral test 36 days before the average kidding date (i.e., Oct-20) to define the social hierarchy among the goats. The behavior test was performed at feeding time (i.e., 08:00, 13:00, and 17:00) for 60 min (180 min per day) for 7 days, as shown in Figure 1. The primary behavioral interactions exerted and documented among goats were hitting, threatening, pushing, chasing, fleeing, and evading (Table 1). In this regard, the observed agonistic interactions between two individuals involved an instigator and a victim, considering the physical displacement of an animal regardless of whether physical contact occurred. With the information obtained from the agonistic interactions, that is, the result of winning or losing, a success rate index (SI) was calculated considering the following formula:SI = number of cases won/(number of cases won + number of cases lost)

The social rank of the 15 pregnant goats was individually classified as either high (HSR) or low (LSR): seven low-rank goats (LSR; IE 0 to 0.49) and eight high-rank goats (HSR; IE 0.5 to 1). Subsequently, regarding the doe, the response variables live weight (LWM, kg), body condition (BCS, units), thoracic diameter (TD, cm), thoracic perimeter (TP, cm), and serum glucose content (GLU, mg dL^−1^) were evaluated. Regarding kids, live weight at birth (LWGK, kg) and litter size were registered.

### 2.4. Quantification of the Response Variables According to Maternal Social Rank

An electronic scale with a capacity of 250 kg and a precision of 50 g (Torrey 110 v/220 v, Digital Industrial Scale, Guadalajara, Mexico) was used to determine live weight. The live weight of the 15 pregnant goats was recorded at 36 d before kidding and later on the kidding day, as well as on days 7, 14, and 21 postpartum. Goats were weighed in the morning before feeding. Body condition score (BCS) was considered an indicator of the level of body reserves present in an animal and, therefore, an indirect marker of metabolic status. The BCS indicates the amount of lipid (i.e., fat) and protein (i.e., muscle) reserves available to the animal for maintenance, reproduction, and production. The BCS is an essential record for producers to optimize production (i.e., meat and/or milk) and define feeding, reproduction, and animal welfare strategies. BCS quantification considered the technique of palpation of the lumbar region [21], quantifying the level of muscle and fat tissue in the animal, using a scale from 1 to 4 (i.e., 1—skinny; 4—fat); specialized technicians carried out this activity.

Regarding the morphometric variables, the thoracic diameter and perimeter were recorded during four consecutive weeks, starting 30 days before kidding. The thoracic diameter was measured with an adjustable zoonometer, and the thoracic perimeter was measured with calibrated tape [22]. The thoracic diameter (TD, cm) was determined by measuring from the dorsal point to the lower sternal region, and the thoracic perimeter TP, cm) was determined by measuring from the lowest dorsal point of the interscapular region towards the lower sternal region to return to the starting point as in Figure 2.

### 2.5. Maternal Serum Glucose Levels and Recording of the Pre-Weaning Growth of the Offspring

A blood sample was collected by jugular venipuncture to quantify serum glucose concentrations (Accu Check Sensor Comfort, Roche, Mexico City, Mexico) with a reliability of 95%. This variable was registered five days before kidding, the day of kidding, and two days post-kidding. Regarding the offspring, the birth weight was registered after parturition, after the mother cleaned it, and before ingesting the colostrum. The live weight of the offspring was also recorded at 7, 14, and 21 days after kidding. The kids’ weight was recorded in the afternoon, ensuring that the kids had avoided suckling for 10 h; a scale with a capacity of 10 kg and a precision of 25 g was used.

### 2.6. Statistical Analyses

The mixed model procedure (i.e., PROC MIXED) was used, analyzing repeated measures across time to evaluate the variables of live weight of the mother (LWM, kg) and kid (LWGK, kg), body condition (BCS, units); the morphological variables of the mother included thoracic diameter (TD, cm) and thoracic perimeter (TP, cm), as well as serum glucose (GLU, mg dL). In all analyses, either the doe or the kid was the experimental unit; the fixed effects of social rank (i.e., LSR and HSR) and sampling day (i.e., time) were evaluated for repeated measures across time; time was considered as a repeated measure while the doe classified by her social rank was considered the repeated subject. In addition, a one-way ANOVA was developed to evaluate the variables according to social rank (i.e., LSR and HS) in crossbred dairy goats (Alpine–Saanen–Nubian × Criollo; *n* = 15).

Since no differences occurred between social rank regarding litter size (i.e., prolificacy), this variable was not considered in the final statistical models. Least-square mean and standard errors for each social rank class, sampling time, and the combination of these two factors were calculated and used to determine multiple comparisons of means via the LSD-Fisher test. All statistical analyses were performed using the procedures of the statistical package SAS version 9.2; a significant difference between means was set at *p* < 0.05.

## 3. Results

### 3.1. Social Rank, Time, and Doe’s Live Weight: Simple Effects

The fixed effects of social rank (i.e., LSR and HS) and time (i.e., five dates) upon the response variable live weight of the mother (LWM) are shown in Table 2. The LSR group presented the lowest value (*p* < 0.05), with an average of 49.3 kg, concerning the average observed in the HSR group (i.e., 55.7 kg). When performing an analysis of the trends of the LWM across time, a recovery of the live weight was observed in the third postpartum week concerning the average weight that the goats showed at the beginning of gestation.

### 3.2. Social Rank, Time, and Doe’s Live Weight Across Time: Interaction Effects

The LWM variable was affected (*p* < 0.05) by the interaction of social rank (i.e., LSR and HSR) and time (i.e., five dates), presented in Table 3. When comparing the minimum and maximum values of the LWM recorded during the experimental period, the highest values either at the beginning (*p* < 0.05; 37.90 vs. 42.9 kg) or at the end of the experiment (i.e., 61.80 vs. 73.40 kg) favored the HSR group. The percentage differences represented 11.3% and 14.4% at the beginning and end of the experiment, always favoring the HSR group.

### 3.3. Social Rank, Time, and Doe’s Body Condition Score Across Time: Interaction Effects

The response variable maternal BCS favored the HSR group (*p* < 0.05) compared to the LSR, either at the onset or the end of the experimental period; moreover, a social rank × time interaction effect was observed (Table 4). The difference between the initial and final BCS average was 0.57 units for the LSR group (2.29 initial and 1.71 final). In contrast, corresponding values of 0.69 were observed in the HSR group, with ranges of 2.44 (initial) to 1.75 (final), favoring the HSR group. When analyzing the simple effects of social rank and time on the BCS, no differences occurred between social ranks, although differences did occur concerning the initial and final BCS; an average reduction of 2.3 vs. 1.8 units occurred, which denoted a drop close to 30% (*p* < 0.05).

### 3.4. Social Rank, Time, and Doe’s Morphometry: Simple Effects

The morphometric maternal variables evaluated considered the thoracic diameter (TD, cm) and the thoracic perimeter (TP, cm). Table 5 shows the values observed for these morphometric variables considering the simple effects of social rank (i.e., LSR and HSR) and time (i.e., four times). While the quantification of the morphometric variables favored the HSR goats, it was observed that these morphometric variables showed an increase in their quantitative values as the kidding date approached.

### 3.5. Social Rank, Time, and Doe’s Morphometry Across Time: Interaction Effects

Once the simple effects on the morphometric variables were analyzed, a second analysis confirmed an interaction effect of social rank (i.e., LSR and HSR) by time (i.e., four dates). In general, the performance of these morphometric variables favored (*p* < 0.05) the HSR group, as shown in Table 6. This interaction presented a positive trend in both social hierarchies. However, the values over time proportionally favored the HSR group.

### 3.6. Social Rank, Time, and Doe’s Serum Glucose Concentration Across Time: Interaction Effects

An interaction (*p* < 0.01) of social rank (i.e., LSR and HSR) and time (i.e., three dates) occurred concerning the variable maternal serum glucose (Table 7). In fact, throughout the experimental period, increases in prepartum, partum, and postpartum glucose levels of 157% and 224% were observed for the LSR and HSR, respectively. However, these levels normalized on subsequent days, considering the mother’s initial and final blood glucose readings, where the percentage increase was 9% and 11% for the LSR and HSR groups, respectively.

### 3.7. Maternal Social Rank, Time, and Kid’s Pre-Weaning Live Weight Across Time: Interaction Effects

The kid’s live weight was affected by the interaction of social rank × time (*p* < 0.05; Table 8). While pre-weaning weights did not differ (*p* > 0.05) between kids from the LSR and HSR groups on days 0, 7, and 14 concerning birth, on d21, live weight favored the LSR kids. Interestingly, however, 66.6% of kids were male and 3% heavier at birth than females. In this regard, while the female kids of the LSR group were 208 g heavier than those of the HSR group, this trend was also observed in male kids since the males of the LSR group were 221 g heavier than those of the HSR group.

## 4. Discussion

The hypothesis raised at the beginning of our research proposed that a high maternal social rank generates higher physio-morphometric values in the mother, promoting higher birth weights and augmented pre-weaning offspring growth from the HSR goats. Our results suggest that HSR goats articulated physiological and morphometric responses to achieve better and greater access to food, increasing live weight values. Still, these physio-ethological advantages were not assumed concerning the pre-weaning growth of their offspring. Our results agree with other reports, indicating that goats are a gregarious species that present interactions between dominant and subordinate animals, translating into complex social interactions. In general, high-ranked animals show better productive and reproductive performance derived from exercising priority access to more and better resources [4,10,11,23,24,25,26,27,28,29,30,31,32].

Indeed, social hierarchies determine unequal access to different resources; goats belong to this group of animals, which naturally have a complex social structure defined by a social hierarchy preserved by agonistic and affiliative behaviors. A stable social environment provides goats and sheep with the ideal conditions to adapt to stressful environments through social learning [1,2,3]. However, the information reported on the importance of social hierarchy is not sufficient, abundant, or recent [33,34,35]. For example, social behaviors of licking and grooming described in herbivores contribute to individual recognition, improve the maintenance of social relations, and favor group cohesion, generating positive affective states [36]. On the other hand, including individuals unknown to the group alters the herd’s social structure and may modify an already-established hierarchical social structure [37,38].

Even under intensive and semi-intensive production schemes, social hierarchy is present, generating continuous competition among the members of the herd and, on many occasions, causing adverse results, mainly related to the spatial distribution of the feeders and, therefore, access to quantity and quality of food, where low-social-rank animals are habitually disturbed by high-social-rank animals. Likewise, let us consider the high food selectivity shown by goats. This feeding behavior favors goats with a high social hierarchy and may exert an enhanced diet intake with higher chemical quality than low-social-rank peers [39,40]. In addition to social rank, food consumption is also affected by other factors, such as the season of the year, environmental temperature, and time of day, among other factors [3,41]. A positive relationship between morphological values and production levels has been reported in goats.

In contrast, over time, these morphological and productive values were positively related to other variables, such as LW and BCS [4,42,43,44]. In our study, the LWM and BCS variables showed higher values at the experiment’s start than at the end. This could be related to the fact that our research period included parturition, a point at which, naturally, does lose substantial energy reserves to ensure the production of colostrum and milk to feed their newborns.

The TD and TP variables showed values with a positive relationship in the social rank by time interaction. They expressed significantly higher values than the initial values at the experiment’s end. Also, a negative relationship was observed between social rank and the amount of serum glucose, that is, goats with LSR showed slightly higher amounts of glucose, except for the kidding day, which is inconsistent with other investigations [11,45]. In this context, such a high glucose level on the partum day in the HSR group could be related to a better LWM and BCS, a scenario previously reported [4,27,28]. Around parturition, a higher social rank generates an increased LWM, which has been related to greater and better access to colostrum by the offspring delivered by HSR goats [3,46,47,48,49,50,51,52]. Interestingly, this scenario was inconsistent with the results obtained in this study. However, colostrogenesis and the process of galactopoiesis were not considered in this research; future research should address this.

## 5. Conclusions

According to our results, even when goats with a high social rank showed higher values in prepartum, partum, and postpartum weights and morphometric values, no differences occurred concerning the maternal serum glucose values, litter size, individual weights of kids, or total litter weights from the kidding-to-weaning period. Our results suggest that although the HSR goats articulated physiological and morphometric responses to achieve better and greater access to food, increasing prepartum and postpartum live weight and zoometric values, such advantages were not reflected in a higher level of productivity neither regarding litter size nor the kid or total offspring weights during the birth-to-weaning period. Such research outcomes are interesting in that it seems that, despite the corporal, morphological, and metabolic increases observed in the HSR goats, such physiological and ethological advantages were not diverted toward better litter size or pre-weaning growth rates; thus, both high- and low-social-rank goats assured the survival of their kids.

## Figures and Tables

**Figure 1 animals-15-01100-f001:**
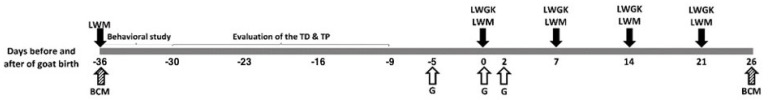
Schematic representation of the study protocol, indicating the main actions carried out throughout the research period. LWM: live weight of the mother; BCM: body condition score of the mother; LWGK: live weight of the kids; TD: thoracic diameter; TP: thoracic perimeter; G: serum glucose.

**Figure 2 animals-15-01100-f002:**
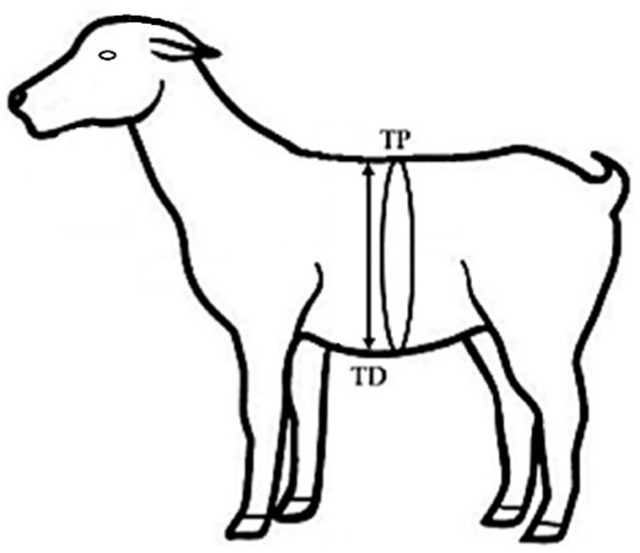
Morphological variables evaluated and their body reference points: thoracic perimeter (TP, cm) and thoracic diameter (TD, cm) from crossbred dairy goats (Alpine–Saanen–Nubian × Criollo; *n* = 15) in northern Mexico (25° N).

**Table 1 animals-15-01100-t001:** Primary behavioral interactions.

Behavior	Description
Hitting	When an individual hits another individual with its head/horns
Threatening	When an individual makes a threatening movement with its head/horns or its trunk and tries to hit another individual
Pushing	When an individual pushes another with its body but without hitting and displaces it from a certain place
Chasing	When an individual chases another around the pen
Fleeing	When an individual moves rapidly away from another
Evading	When an individual avoids another’s threats or presence

**Table 2 animals-15-01100-t002:** Effect of social rank on live weight.

Variable	Social Rank			Days Regarding Kidding Date ^1^					
	LSR	HSR	s.e.	−36	0	+07	+14	+21	s.e. ^3^
LWM (kg)	49.3 ^b 2^	55.7 ^a^	1.8	52.6 ^ab^	58.8 ^a^	51.5 ^ab^	50.2 ^ab^	49.1 ^b^	2.9

Least-square means + s.e. for live weight of does (LWM, kg) as affected by either social rank (i.e., LSR and HSR) or time (i.e., five dates) in crossbred (Alpine–Saanen–Nubian × Criollo; *n* = 15) dairy goats in northern Mexico (25° N). ^1^ Expected kidding date, 25 November. ^2 ab^ Least-square means without a common superscript within response variables differ significantly (*p* < 0.05). ^3^ Standard error.

**Table 3 animals-15-01100-t003:** Social rank, time, and live weight interaction.

Days Regarding Kidding Date ^1^											
	−36		0		07		14		21		s.e. ^2^
Variable	LSR	HSR	LSR	HSR	LSR	HSR	LSR	HSR	LSR	HSR	
LWM (kg)	49.5 ^cdef^	55.8 ^b^	55.7 ^b^	62.0 ^a^	48.5 ^def^	54.6 ^bc^	47.3 ^ef^	53.2 ^bcd^	45.3 ^f^	52.9 ^bcde^	2.0

Least-square means ± s.e. of the doe’s live weight (LWM, kg) according to the interaction of social rank (i.e., LSR and HSR) and time (i.e., five dates) in crossbred (Alpine–Saanen–Nubian × Criollo; *n* = 15) dairy goats in northern Mexico (25° N). ^1^ Expected kidding date, November 25. ^abcdef^ Least-square means without a common superscript within response variables differ (*p* < 0.05). ^2^ Standard error.

**Table 4 animals-15-01100-t004:** Social rank, time, and body condition score interaction.

Variable	Initial ^1^		Final ^2^	
	LSR	HSR	LSR	HSR
BCS (units)	2.2 ± 0.14 ^a^	2.4 ± 0.13 ^a^	1.8 ± 0.14 ^b^	1.8 ± 0.13 ^b^

Least-square means ± s.e. for the doe’s body condition score (BCS, units) according to the initial and final dates according to social rank (i.e., LSR and HSR) in crossbred (Alpine–Saanen–Nubian × Criollo; *n* = 15) dairy goats in northern Mexico (25° N) ^1^. The expected kidding date was November 25. ^1^ Initial: 36 days prior kidding. ^2^ Final: 25 days post-kidding. ^ab^ Least-square means without a common superscript within responses differ (*p* < 0.005).

**Table 5 animals-15-01100-t005:** Effect of social rank and time on thoracic diameter and perimeter.

Variable	Social Rank			Days Regarding Kidding Date ^1^				
	LSR	HSR	s.e.	−30	−23	−16	−09	s.e. ^3^
TD (cm)	33.9 ^b 2^	35.7 ^a^	0.6	31.6 ^b^	34.9	36.0 ^a^	36.8 ^a^	0.8
TP (cm)	108.4 ^a^	112.6 ^a^	1.4	107.2 ^b^	109.0 ^ab^	112.1 ^ab^	114.4 ^a^	2.3

Least-square means ± s.e. for doe’s thoracic diameter (TD, cm) and thoracic perimeter (TP, cm), according to social rank (i.e., LSR and HSR) and time (i.e., four dates) in crossbred (Alpine–Saanen–Nubian × Criollo; *n* = 15) dairy goats in northern Mexico (25° N). ^1^ Expected kidding date, November 25. ^2 ab^ Least-square means without a common superscript within response variables differ (*p* < 0.05). ^3^ Standard error.

**Table 6 animals-15-01100-t006:** Thoracic diameter and perimeter interaction with social rank.

Variable	Days Regarding Kidding Date ^1^								
	−30		−23		−16		−09		s.e ^3^
	LSR	HSR	LSR	HSR	LSR	HSR	LSR	HSR	
TD (cm)	30.8 ^e 2^	32.3 ^de^	33.8 ^cd^	35.9 ^ab^	35.1 ^c^	36.9 ^ab^	35.1 ^c^	36.9 ^ab^	0.6
TP (cm)	105.1 ^e^	109.0 ^cd^	106.7 ^de^	111.2 ^bc^	109.5 ^bc^	114.3 ^ab^	109.5 ^cde^	114.3 ^ab^	1.7

Least-square means ± s.e. for the interaction of social rank (i.e., LSR and HSR) and time (i.e., four dates) for the thoracic diameter (TD, cm) and thoracic perimeter (TP, cm) of the mother in crossbred (Alpine–Saanen–Nubian × Criollo; *n* = 15) dairy goats in northern Mexico (25° N). ^1^ Expected kidding date, November 25. ^2 abcde^ Least-square means without a common superscript within response variables differ (*p* < 0.05). ^3^ Standard error.

**Table 7 animals-15-01100-t007:** Effect of social rank on glucose levels.

Variable	Days Regarding Kidding Date ^1^						
	−05		0		+02		s.e ^2^
	LSR	HSR	LSR	HSR	LSR	HSR	
GLU (mg dL^−1^)	39.4 ^b^	37.1 ^b^	101.5 ^a^	116.0 ^a^	43.1 ^b^	41.5 ^b^	3.3

Least-square means ± s.e. for doe’s serum glucose levels (GLU, mg dL-1) as affected by the social rank (i.e., LSR and HSR) and time (i.e., three dates) interactions in crossbred (Alpine–Saanen–Nubian × Criollo; *n* = 15) dairy goats in northern Mexico (25° N). ^1^ Expected kidding date: November 25. ^ab^ Least-square means without a common superscript within the response variable differ (*p* < 0.05). ^2^ Standard error.

**Table 8 animals-15-01100-t008:** Interaction between the kid’s pre-weaning live weight and the mother’s social rank.

Variable	Days Regarding Kidding Date ^1^								
	0		07		14		21		s.e ^2^
	LSR	HSR	LSR	HSR	LSR	HSR	LSR	HSR	
LWGK (kg)	3.5 ^e^	3.3 ^e^	4.8 ^d^	4.5 ^d^	6.1 ^c^	5.7 ^c^	7.6 ^a^	6.8 ^b^	0.2

Least-square means ± s.e. for the pre-weaning live weight of kids (LWGK, kg) as affected by the interaction of social rank of their mothers (i.e., LSR and HSR) and time (i.e., three dates) in crossbred (Alpine–Saanen–Nubian × Criollo; *n* = 15) dairy goats in northern Mexico (25° N). ^1^ Expected kidding date, November 25. ^abcde^ Least-square means without a common superscript within the response variable differ (*p* < 0.05). ^2^ Standard error.

## Data Availability

The datasets used in this research are available from the corresponding author upon reasonable request.

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
