# Peer review of "Maternal Social Hierarchy, Morphometric Traits, Live Weight, and Metabolic Status as Related to the Offspring Pre-Weaning Growth in Crossbred Dairy Goats"

_animals, 2025, doi:10.3390/ani15081100_

Round 1
Reviewer 1 Report
Comments and Suggestions for Authors
General comments
This study mainly aimed to evaluate the effect of maternal social hierarchy on several traits of does and offsprings. The subject has relevance for goat production. The study design is adequate for the purposes. Overall, the manuscript was well structured and presented. I suggest inserting at least 2 paragraphs in the introduction to separate ideas. Overall,l M&M is sufficiently informative to ensure the replicability of the experiment, but more information about feeding/nutritional management of kids is required. The litter size was not different between groups and not considered in the statistical models. Nonetheless, the sample size is low, and a relevant question arises: what is the absolute percentage of singletons and twins? Can this proportion affect LWGK? Note that the absolute values at different points in time are always higher in LSR than HSR group (i.e., we are talking about statistical power). This can justify at least partially your discussion in L314-320? The results are fully supported by 7 independent tables. The findings are sufficiently discussed regarding the published literature. The conclusions are full supported by the results. No major issues were found.
Specific comments:
L99-101: This is a repetition.
L109: It was preferable to present a mean and SE for day 36?
L117,121: 135/139?
133-136: Morning?
L166: Decimals are preferred instead of centesimals.
Comments on the Quality of English LanguageNone.
Author Response
First, thank you very much for the comments. They have been very valuable in increasing the quality of the manuscript.
Comment 1: This study mainly aimed to evaluate the effect of maternal social hierarchy on several traits of does and offsprings. The subject has relevance for goat production. The study design is adequate for the purposes. Overall, the manuscript was well structured and presented. I suggest inserting at least 2 paragraphs in the introduction to separate ideas.
R: The introduction was separated into paragraphs
Comment 2: The litter size was not different between groups and not considered in the statistical models. Nonetheless, the sample size is low, and a relevant question arises: what is the absolute percentage of singletons and twins?
R: The sample size is small because, when selecting the animals for the study, the social rank evaluation is done in the whole herd, and most animals are in the middle, we only selected animals in the high and low social ranks. The percentage of prolificacy was as follows: 4/13(30.8%) had twins, 1/13 (7.7%) had triplets, and 8/13 (61.5%) were singletons.
Comment 3. Can this proportion affect LWGK? Note that the absolute values at different points in time are always higher in LSR than HSR group (i.e., we are talking about statistical power). This can justify at least partially your discussion in L314-320?
R: All kids' average weight at birth was 3.6 kg, but prolificity was higher for HSR (1.6 vs 1). This could partially explain the lower average weight for LSR kids
Comment 4. The results are fully supported by 7 independent tables. The findings are sufficiently discussed regarding the published literature. The conclusions are full supported by the results. No major issues were found.
R: Thank you
Comment 5: L99-101: This is a repetition.
R: removed
Comment 6: L109: It was preferable to present a mean and SE for day 36?
R: due to management practices, data was collected on day 36
Comment 7: L117,121: 135/139?
R: removed
Comment 8: L166: Decimals are preferred instead of centesimals.
R: this was corrected
Reviewer 2 Report
Comments and Suggestions for Authors
General comments of the reviewer:
The manuscript is dealing with a relevant topic, but it seems that there was not enough time to refine the text. There are many repetitions in the text and formal errors, as well. The methodology should be refined (marked in the text) and the presentation of the results should be upgraded.
Please follow the formal requirements of the Journal.
The detailed comments and recommendations are in the attached version of the manuscript.

Generally, the English is good, but there are grammatical errors in the text which should be checked by a native speaker.
Author Response
First of all, thank you for all the comments, they have been vary valuable for increasing the quality of the manuscript.
Comment 1. L18: Please move this sentence to the Introduction part.
R= It has been moved to L45-47.
Comment 2. L20: The possible effect of maternal social rank (SR) was evaluated in this trial concerning…..
R: SR was added
Comment 3. L32: What do you mean by that exactly?
R: this was removed
Comment 4. L69: Which year?
R: 2023
Comment 5. L74: n=15
R: done
Comment 6. L74: What was the management?
R: changed to “managed under semi-intensive conditions where goats grazed from !0:00 to 18:00 h in different grazing sites on a path of approximately 6 to 8 km”
Comment 7. L83: Can you refine this sentence?
R: the sentence was changed, it now reads “The behavior test was performed at feeding time (i.e., 08:00, 13:00, and 17:00) for 60 min (180 min day) for 7 days as in Figure 1.”
Comment 8. L85: It is recommended to described the interactions in details (e.g. in a table - what do you mean exactly on each interaction).
R: a table was added giving a detailed description of each interaction (Table 1) al the other tables were renamed accordingly.
Comment 9. L97: This figure needs to be placed more precisely. The abbreviations needs to be described below.
R: done
Comment 10. L100: It is repetition! You already wrote about that.
R: removed
Comment 11. L105: It is also repeted, not needed here again.
R: removed
Comment 12. L121: What is this number?
R: removed
Comment 13. L136: What was the litter size of each animal? How did you compare the does with different litter sizes?
R: Average prolificacy was 1.5
Comment 14. L136: parturition
R: done
Comment 15. L154: OK, but what do you thing about the relation between litter size and LWM of does?
R: HSR does have a higher prolificacy than LSR does (1.6 vs 1); however, LWGK had no statistical differences.
Comment 16. L167: Where is this number exactly in Table 1?
R: it was corrected to 55.7
Comment 17. L172: To many information in the titles! Please shorten the titles and put all explanations below the tables.
R: done
Comment 18. L192: Please use the same name consistently throughout the manuscript (doe is recommended).
R: Dam was replaced by doe
Comment 19. L257: kids
R: done
Comment 20. L304: ...and please don't forget what the parturition means! What was the litter size of each doe? Unfortunately, this very important information is not included in the manuscript.
R: Average prolificacy was 1.5, with HSR does having more twin births than LSR does (1.6 vs 1), but the LWGK was not affected.
Comment 21. L317: Where do you mention the results of your trial regarding to this sentence?
R: It is not, the sentence was replaced with “Interestingly, this scenario was inconsistent with the results obtained in this study. However, colostrogenesis and the process of galactopoiesis were not considered in this research; future research should address this.”
Comment 22. L325: Where are the results regarding to that?
R: same as above
Round 2
Reviewer 2 Report
Comments and Suggestions for Authors
Authors have accepted the reviewer's recommendations and modified that parts wich have been criticized.